# Reheated Gradient-based Discrete Sampling for Combinatorial Optimization

**Muheng Li** *                                                          *muheng.li@mail.utoronto.ca*
*Department of Statistical Sciences*
*University of Toronto*

**Ruqi Zhang**                                                               *ruqiz@purdue.edu*
*Department of Computer Science*
*Purdue University*

**Reviewed on OpenReview:** *https://openreview.net/forum?id=uPCvfyr2KP*

## Abstract

Recently, gradient-based discrete sampling has emerged as a highly efficient, general-purpose solver for various combinatorial optimization (CO) problems, achieving performance comparable to or surpassing the popular data-driven approaches. However, we identify a critical issue in these methods, which we term "wandering in contours". This behavior refers to sampling new different solutions that share very similar objective values for a long time, leading to computational inefficiency and suboptimal exploration of potential solutions. In this paper, we introduce a novel reheating mechanism inspired by the concept of critical temperature and specific heat in physics, aimed at overcoming this limitation. Empirically, our method demonstrates superiority over existing sampling-based and data-driven algorithms across a diverse array of CO problems.

## 1 Introduction

Combinatorial optimization (CO) refers to the problem that seeks the optimal solution from a finite feasible set. This type of optimization is prevalent in numerous applications such as scheduling, network design, resource allocation, and logistics. CO problems are inherently complex due to the huge number of potential combinations, making the search for the optimal solution challenging.

To solve CO problems, various methods are developed. Exact methods like Branch-and-Bound (Land & Doig, 2010) guarantee optimal solutions but are computationally intensive for large problems. Heuristic approaches, such as greedy algorithms (DeVore & Temlyakov, 1996), offer quick solutions but without an optimality guarantee. Integer linear programming (Schrijver, 1998) is effective for formulating problems mathematically, yet it struggles with scalability. Recently, data-driven approaches using machine learning (Li et al., 2018b; Gasse et al., 2019; Khalil et al., 2017; Nair et al., 2020; Karalias & Loukas, 2020) have seen rapid growth. However, they demand significant data and computational power and cannot be used as off-the-shelf solvers for problems that differ from the training data.

Recently, sampling approaches have regained popularity in addressing CO problems. Simulated annealing (SA) (Kirkpatrick et al., 1983), a longstanding sampling method, has been widely applied in the CO field for several decades. However, these methods are often criticized for their inefficiency in high-dimensional spaces, primarily due to their random walk behavior during the exploration (Delahaye et al., 2019; Nourani & Andresen, 1998; Haddock & Mittenthal, 1992). Leveraging recent advancements in gradient-based discrete sampling (Grathwohl et al., 2021; Sun et al., 2021; 2023a; Zhang et al., 2022), Sun et al. (2023b) demonstrates that sampling methods can serve as general solvers and achieve a more favorable balance between speed and

---

*Work done during an internship at Purdue University.

solution quality compared to data-driven approaches for many CO problems, sometimes even outperforming commercial solvers and specialized algorithms.

In this paper, we pinpoint a critical limitation, termed *wandering in contours*, in current gradient-based sampling for CO. We observe that the sampling methods remain stuck in sub-optimal contours, defined as a set of diverse solutions with identical or very similar objective values, for a prolonged duration due to reliance on gradient information. This phenomenon severely harms both efficiency and solution quality. To tackle this problem, we introduce a novel reheating mechanism inspired by the concept of critical temperature and specific heat in physics. Our approach involves resetting the temperature to a theoretically informed threshold once the algorithm begins wandering, thus facilitating more effective exploration. We summarize our contributions as follows:

- We identify a key issue in gradient-based sampling for combinatorial optimization (CO), termed *wandering in contours*: algorithms yield distinct solutions, yet with nearly identical objective values for extended periods due to using gradient and low temperatures.

- We then propose a reheat mechanism where we reset the temperature to the critical temperature once the algorithm begins wandering, optimizing resource use and improving solution discovery.

- Through extensive experiments on various CO problems, we demonstrate that our method significantly advances over previous sampling approaches. The code is available at `https://github.com/PotatoJnny/ReSCO`.

## 2 Related Work

**Gradient-based Discrete Sampling**  Sampling in discrete domains has historically been challenging, with Gibbs sampling (Geman & Geman, 1984) being a longstanding primary method. A significant improvement came with the introduction of locally balanced proposals by Zanella (2019), leveraging local solution information for sampling in discrete neighborhoods. Gradient-based discrete samplers are derived by using gradient approximation (Grathwohl et al., 2021), which become the most common implementations of locally balanced samplers. Later, Sun et al. (2021) proposes PAS to broaden the discrete sampling neighborhood, and Discrete Langevin developed by Zhang et al. (2022) enables parallel updates in each step. Many discrete samplers are developed based on variants of locally balanced proposals (Rhodes & Gutmann, 2022; Sun et al., 2022; 2023a; Xiang et al., 2023; Pynadath et al., 2024)

**Sampling for Combinatorial Optimization**  Sampling methods were once popular for solving combinatorial optimization (CO) problems, among which the most famous algorithm is simulated annealing (SA) Kirkpatrick et al. (1983). SA has been successfully applied to a wide array of CO problems, including TSP (Kirkpatrick et al., 1983), scheduling (Van Laarhoven et al., 1992), and vehicle routing (Osman, 1993). Though SA theoretically guarantees convergence to the global optimum under certain conditions, its use of random walk leads to slow convergence in high-dimensional discrete problems. Addressing this, Sun et al. (2023a) introduces an improved algorithm by integrating the gradient-based discrete sampler into the SA framework, achieving superior solutions for complex CO problems more efficiently than many learning-based approaches.

## 3 Preliminaries

### 3.1 Combinatorial Optimization

We consider combinatorial optimization problems with the form:

$$\min_{x \in \Theta} u(x), \quad \text{s.t.} \quad v(x) = 0 \tag{1}$$

where $\Theta$ is a $d$ dimensional finite solution space. We assume $\forall x \in \Theta$, the objective value $u(x) \geq 0$ and $v(x) \leq 0$. By introducing a penalty coefficient $\lambda$, we obtain an unconstrained optimization problem from Eq

equation 1 (big enough $\lambda$ will ensure the equivalence of the optimal solutions):

$$\min_{x \in \Theta} f(x) \triangleq u(x) + \lambda v(x). \tag{2}$$

We can view it as an energy-based model, where $x$ is a state and $f(x)$ is the energy of the state.

### 3.2 Simulated Annealing

Simulated annealing (SA) optimizes energy-based models by simulating a physical annealing. It employs a temperature parameter $T$, and states are sampled according to

$$\pi_T(x) \propto \exp(-f(x)/T). \tag{3}$$

Inhomogeneous SA performs a single sampling step at each $T$ while homogeneous SA performs multiple steps. The temperature gradually decreases, and new solutions $x'$ are accepted based on:

$$P_T(x \to x') = \min(1, \exp(-(f(x') - f(x)/T)). \tag{4}$$

During the transition from high to low temperatures, SA narrows the target distribution, shifting from wide exploration to targeted exploitation of low-energy states. SA is theoretically guaranteed to converge to the global optimal state as $T$ goes to 0.

### 3.3 Gradient-based Discrete Samplers

In locally balanced proposal (Zanella, 2019), the update rule at the current state $x$ is given by:

$$q(y, x) \propto g\left(\pi(y)/\pi(x)\right) K(x - y) \tag{5}$$

where $\pi(\cdot)$ is the target distribution over the discrete space $\Theta$, $K$ is a kernel that decides the scale of the proposal, and $g(\cdot)$ is a balancing function. In energy-based models, we have $\pi(x) \propto \exp(-f(x))$, i.e. $\pi_1(x)$ in equation 3. Then,

$$q(y, x) \propto g(\exp(f(x) - f(y))K(x - y) \tag{6}$$

Using the first-order Taylor expansion to approximate $f(x) - f(y)$, we then get the gradient-based discrete sampler. Here, we assume $f$ has a natural continuous relaxation $\tilde{f}$. The gradient of $f$ is defined as $\nabla f(x) = \nabla \tilde{f}(x), x \in \Theta$.

## 4 Pitfall of Gradient-based Discrete Sampling

In Section 4.1, we present an intriguing phenomenon of gradient-based sampling in solving combinatorial optimization problems, which remarkably affects its performance and efficiency. In Section 4.2, we present a theoretical analysis of this phenomenon and propose two explanations for its occurrence.

### 4.1 Wandering in Contours

Gradient-based discrete sampling, when combined with simulated annealing settings, has shown considerable success in addressing large-scale combinatorial optimization (CO) problems (Sun et al., 2023b; Goshvadi et al., 2023), mainly due to its significantly accelerated convergence rate. The use of gradient information, though advantageous for locating good solutions quickly, also undermines

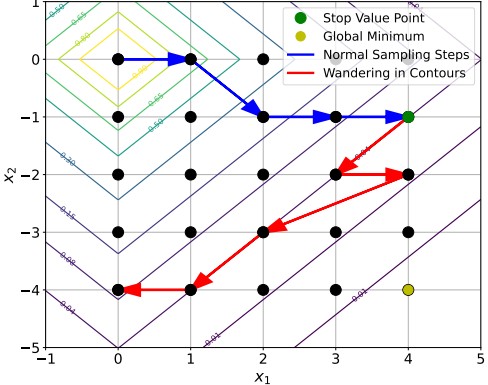

Figure 1: Visualization of the "wandering in contours" phenomenon.

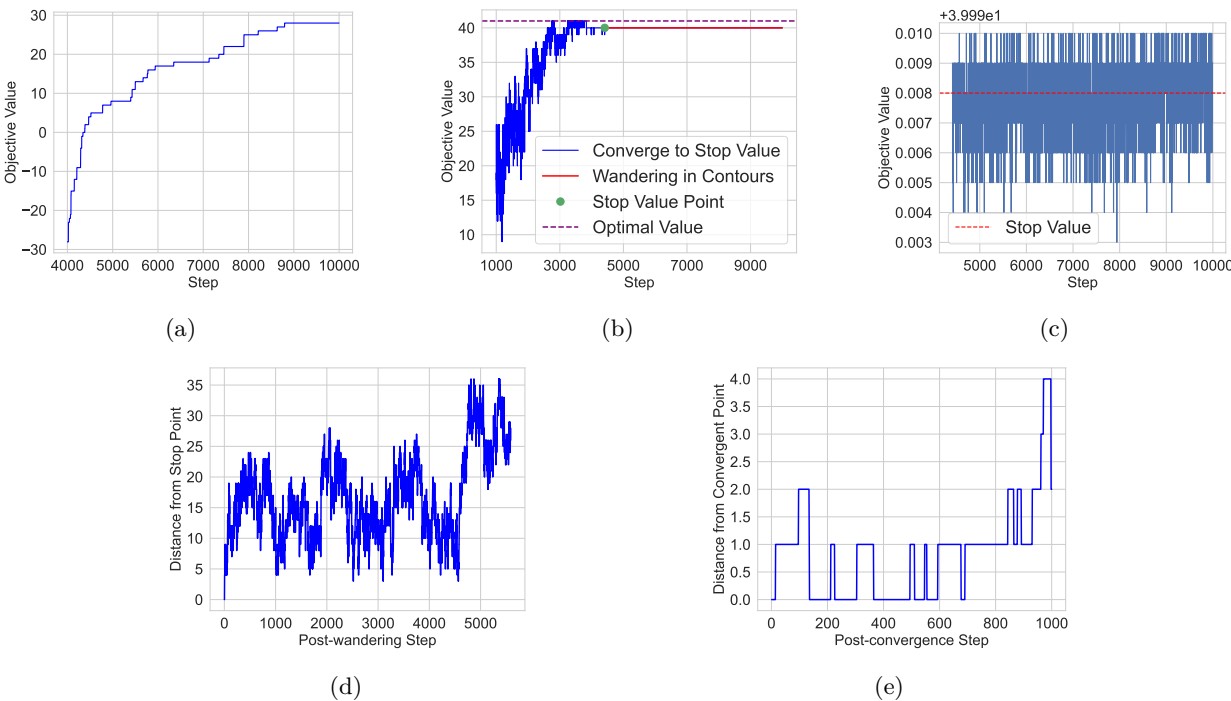

Figure 2: The wandering in contours phenomenon of a gradient-based discrete sampler on a maximum independent set problem. **(a)**: The trajectory of objective values of traditional simulated annealing, where the algorithm updates the value slowly. **(b)**: The trajectory of objective values of gradient-based sampling methods. The algorithm converges rapidly to a stop value solution (highlighted in green) and then stays close to this value. **(c)**: The objective values after the stop point continue to fluctuate but stay very similar to or the same as the stop value. **(d)**: Gradient-based samplers produce solutions with similar objective values but large distances from the stop point. **(e)**: All the newly sampled solutions in traditional simulated annealing are near each other.

the core strength of simulated annealing to explore the solution space effectively. This leads to a peculiar form of trap in the optimization process, which we term *wandering in contours*,

**Wandering in Contours**: *After quickly converging to a sub-optimal stop value, the algorithm begins to sample new **different** solutions sharing objective values very **similar** to, or even the **same** as, the stop value for a long time.*

We observe that "wandering in contours" is prevalent across various gradient-based discrete sampling methods and CO problems. This behavior is visualized in Figure 1, where the algorithm, upon reaching a stop value, appears to wander within certain contour lines without making further progress.

Empirical evidence of this behavior is presented in Figure 2. Here, we use a gradient-based discrete sampler, PAS (Sun et al., 2021), within simulated annealing (SA) framework, for a maximum independent set problem, aiming to find the solution with the highest objective value. Compared to the slowly updating process of traditional SA depicted in Figure 2a, the simulated annealing with PAS rapidly converges to a sub-optimal solution (marked in green) within approximately 4,000 of the total 10,000 steps as shown in Figure 2b. Following this convergence, the algorithm enters a phase of "wandering in contours", where it generates new, different solutions, yet their objective values are very similar to the stop value, as demonstrated in Figure 2c & 2d. This phenomenon is broadly present in various gradient-based discrete samplers, with evidence provided in Appendix A.

**How is "wandering in contours" different from convergence to suboptimal points?** "Wandering in contours" is distinct from being trapped in suboptimal points, a well-known phenomenon in machine

learning and optimization (Zhang et al., 2017). While both involve a lack of significant improvement, "wandering in contours" depicts a phenomenon that the algorithm will sample different solutions, which is just like the algorithm is still forwarding along a road and getting farther away from the beginning solution of "wandering in contours", but all the solutions on this road share similar objective values. This behavior is illustrated in Figure 2d, where the new solutions sampled by the algorithm are getting far away from the stop solution, while they share almost the same objective values.

In contrast, convergence to the suboptimal points signifies that the algorithm has found a region containing the (sub)optimal solution. As mentioned in Cruz & Dorea (1998), convergence indicates that the solutions sampled become increasingly clustered around the suboptimal solution. This clustering behavior can be observed in Figure 2e, which shows the distance between the newly sampled solutions and the convergent solution. It can be clearly seen that the newly sampled solutions are near each other.

### 4.2 What causes "wandering in contours"?

In the previous section, we observed that the sampler tends to "wander in contours" after reaching a certain point, failing to make further progress. This section delves into an analysis of this phenomenon and identifies its underlying causes. We discovered that this behavior primarily stems from two factors: *the misleading gradient information* employed by discrete samplers, and *the low-temperature environment* inherent in the simulated annealing framework. These two factors together significantly diminish the algorithm's ability to escape suboptimal solutions, leading it to persist in the "wandering in contours" pattern.

**Misleading Gradient Information** In gradient-based discrete sampling algorithms, the gradient $\nabla f(x)$ will be incorporated in the proposal to generate the next state. The quality of the proposed state thus highly depends on whether the gradient information is useful.

It is well-established that gradient-based methods in continuous spaces are susceptible to converging to local optima because gradient information can be misleading. This issue becomes even more pronounced in discrete optimization problems, where the gradient utilized by discrete samplers is derived from a natural continuous relaxation of the original discrete domain (as discussed in the preliminaries section). In such relaxed continuous spaces, local minima may not correspond to feasible discrete solutions. Instead, they may be encircled by a cluster of feasible discrete solutions sharing very similar objective values. The misleading gradient will then lead the algorithm to jump around these feasible solutions, causing the wandering in contours behavior.

We take a 1-dimensional discrete optimization problem as an example:

$$\min_{x \in \mathbb{N}} \frac{1}{4}x^4 - \frac{4}{3}x^3 + \frac{15}{8}x^2.$$

The global minimum of the function is located at $x = 0$, as illustrated in Figure 3a. Within the continuous relaxation of the domain $\mathbb{N}$, there exists a local minimum at $x = 2.5$. This local minimum is surrounded by the feasible discrete solutions at $x = 2$ and $x = 3$. The gradient-based discrete samplers will be influenced by gradient information to move towards the local minimum at $x = 2.5$. However, since $x = 2.5$ is not a feasible solution in the discrete domain, the algorithm would ultimately converge to either $x = 2$ or $x = 3$, and then jump to the other directed by the gradient. This behavior can lead to the algorithm oscillating between these two suboptimal solutions.

**Low Temperature Environment** The low-temperature stage is a critical phase in the simulated annealing process, featuring a rapid convergence toward the final solution as the algorithm reduces its random exploration. However, during this phase, the exploration capabilities of gradient-based discrete samplers, enabled by the randomness in the proposal distribution, rapidly diminish, making them less likely to escape from suboptimal solutions.

We take one of the gradient-based discrete sampler DMALA (Zhang et al., 2022) as an example, and the following analysis is also applied to the other gradient-based discrete samplers. The update rule of DMALA at temperature $T$ is:

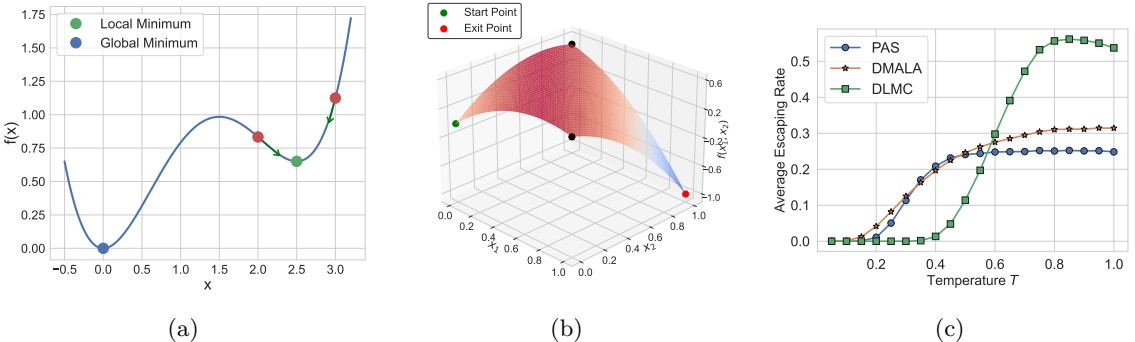

(a)            (b)            (c)

Figure 3: **(a):** Illustration of the discrete optimization landscape for the function $\frac{1}{4}x^4 - \frac{4}{3}x^3 + \frac{15}{8}x^2$. Directed by the gradient information, the gradient-based discrete samplers will oscillate between two discrete solutions 2 and 3 near the local minimum 2.5. **(b):** Visualization of $f(x_1, x_2)$ from equation 7. In the experiment, we start from the local minimum $(0, 0)$ (marked as green) and test whether the discrete samplers can reach the global minimum $(1, 1)$ (marked as red); **(c):** The average escaping rate of three gradient-based discrete samplers, which is defined as the probability of escaping from local minimum $(0, 0)$ to the global minimum $(1, 1)$ by running 20 sampling steps at different temperatures, declines sharply as temperature decreases.

$$p(x_i'|x) = \text{Softmax}\left( \frac{1}{2} \frac{\nabla f(x)_i (x_i' - x_i)}{T} - \frac{(x_i' - x_i)^2}{2\alpha} \right).$$

When $T$ is low, the magnitude of the gradient term increases significantly. As a result, the values inside the softmax function become highly varied for different $x'$. This results in a more concentrated probability distribution after taking the softmax function. Consequently, the selection process becomes more deterministic and less random, as the algorithm strongly favors certain directions. This means that when the temperature is low, once the gradient information is misleading, as discussed before, the gradient-based discrete sampling is more prone to the "wandering in contours" behavior and less likely to escape from it.

Moreover, at a low temperature $T$, the probability of accepting a worse solution $x'$ in simulated annealing is lower, making it harder to jump out of the local optima by hill-climbing, i.e., temporarily accepting worse solutions for eventually reaching superior ones.

We now show the effect of low temperature by a toy example. We consider the following discrete optimization problem:

$$\min_{(x_1, x_2) \in \{0,1\}^2} f(x_1, x_2) = -(x_1 + x_2)(x_1 + x_2 - \frac{3}{2}). \tag{7}$$

This function features a local minimum $(0, 0)$ and a global minimum $(1, 1)$, and hill-climbing is needed for escaping from $(0, 0)$ to $(1, 1)$, as illustrated in Figure 3b.

We employed three distinct gradient-based discrete samplers DMALA (Zhang et al., 2022), DLMC (Sun et al., 2023a), and PAS (Sun et al., 2021) to address this optimization problem under varying constant temperature conditions. Initiating from the local minimum $(0, 0)$, each sampler executed 20 sampling steps at various constant temperatures to assess their ability to transition to the global minimum $(1, 1)$. Repeating 100,000 experiments for each temperature setting allowed us to calculate an average escaping rate, defined as the likelihood of successfully moving from $(0, 0)$ to $(1, 1)$ within 20 steps.

The results, shown in Figure 3c, clearly illustrate a sharp decline in the escaping rate as temperature decreases across all samplers. This trend underscores the critical influence of temperature in gradient-based discrete samplers within the simulated annealing framework, particularly highlighting how lower temperatures significantly reduce the algorithm's ability to escape from suboptimal solutions.

## 5 Gradient-based Sampling with Reheat

To address the challenges outlined in Section 4, we propose a reheat mechanism, which strategically raises the temperature upon detecting the algorithm's wandering. This approach addresses misleading gradient information by flattening the proposal distribution over different candidates at higher temperatures, adding randomness and aiding the escape from suboptimal states. The reheat directly tackles low-temperature limitations by increasing the temperature, facilitating a more dynamic exploration process. Intuitively, reheat works by resetting the temperature to a higher value. It usually contains two key steps: (1) Detect when to reheat. (2) Increase the temperature to a predefined higher value. Below, we present our reheat method for gradient-based discrete samplers in detail.

Our idea of reheating is non-trivial. The reheating mechanism is different from existing stepsize-tuning methods explored in MCMC literature, and a detailed discussion is provided in Appendix C. While simulated annealing with reheating has been proposed before (Abramson et al., 1999; Anagnostopoulos et al., 2006), its usage remains very limited. This is primarily due to the slow convergence rate of the original simulated annealing (SA). Reheat was designed to aid simulated annealing in escaping local optima, but the slow convergence of SA made it challenging to even reach a local optimum. Consequently, the reheat mechanism often had minimal impact on enhancing simulated annealing's performance and, in certain cases, could even degrade it. On the contrary, gradient-based discrete samplers can converge to the local optimum quickly, which re-energizes the reheating strategy. Integrating simulated annealing with gradient-based discrete samplers offers a situation where the reheat mechanism becomes effective.

### 5.1 Detecting When to Reheat

The first critical step of our reheat mechanism is accurately detecting the "wandering in contours" behavior. Considering that the objective values during wandering remain very similar, this behavior can be detected by measuring the variation between the objective values.

For this purpose, we set a value threshold, $\epsilon$, and a wandering length threshold, $N$. These parameters are used to assess the changes in the objective values $f(x_t)$ at each step $t$. Specifically, we monitor the absolute differences in consecutive objective values, and if these differences fall below $\epsilon$ for $N$ consecutive steps, we infer that the algorithm is exhibiting "wandering in contours". Formally, the algorithm is considered to be wandering in contours at step $t$ if:

$$|f(x_{t-i}) - f(x_{t-i-1})| < \epsilon, \quad \forall i = 0, 1, \dots, N-1. \tag{8}$$

Hyperparameters $\epsilon$ and $N$ can be tailored to specific problems and the expected variability in the objective values. Generally, a smaller $\epsilon$ makes the detection more sensitive to minor changes, and a smaller $N$ reduces tolerance for spending time on similar values. We studied their practical effect in Section 6.3.3.

### 5.2 Selecting the Reset Temperature

After detecting "wandering in contours", choosing an appropriate reset temperature is critical. If the reset temperature is too low, then the sampler will not have enough exploration ability to accept worse solutions temporarily and diminish the influence of gradient information. If the reset temperature is too high, then the sampler will act like an inefficient random walk for a long time.

#### 5.2.1 Critical Temperature

To identify the optimal reset temperature, we utilize the concept of *critical temperature* $T_c$ in physical annealing (Landau & Lifshitz, 1976). This temperature marks the phase transition from disordered to ordered states, fundamentally changing physical properties. Similar phase transitions have been observed in simulated annealing (Kirkpatrick et al., 1983). Above $T_c$, the sampler's search is random and broad. Below $T_c$, the sampler is selective, with a higher tendency to reject worse solutions. It is also observed that the search will be trapped in a single valley below $T_c$, making it hard to jump out of the wandering in contours (Strobl & Barker, 2016). Hence, $T_c$ serves as a critical balance between exploration and exploitation.

The efficiency of simulated annealing around $T_c$ has been reported in numerous studies. Kirkpatrick et al. (1983) demonstrated that during phase transition, the search becomes more efficient. Strobl & Barker (2016) showed that when using simulated annealing to solve phylogeny reconstruction, the search is constrained to a small valley of the search space when the temperature is below $T_c$. The search efficiency around $T_c$ can also be seen in Figure 2 of (Cai & Ma, 2010b).

While some algorithms (Basu & Frazer, 1990; Cai & Ma, 2010a) have used critical temperature $T_c$ to design the initial and final temperatures, its use for reheating is less common. Abramson et al. (1999) proposes a reheating strategy tied to function cost that implicitly involves $T_c$. However, this method lacks theoretical support and is sensitive to hyperparameters. Instead, our approach directly reheats to the critical temperature, which injects just the right amount of stochastic energy back into the system, enabling it to escape "wandering in contours" without incurring excessive randomness of a high-temperature regime.

### 5.2.2 Specific Heat

Determining the critical temperature in simulated annealing requires identifying the phase transition point during optimization, which is characterized by peaks in *specific heat* (Kirkpatrick et al., 1983; Strobl & Barker, 2016). In statistical physics, the system's energy at temperature $T$, denoted as $E(T)$, adheres to the Boltzmann distribution. Thus, the expected energy of a system can be seen as a function of the temperature, $\mathbb{E}[E(T)]$. The specific heat at temperature $T$, denoted as $C_T$, is traditionally defined in thermodynamics as the rate of change of the expected energy $\mathbb{E}[E(T)]$ to temperature (Aarts et al., 1987), given by $C_T = \frac{\partial \mathbb{E}[E(T)]}{\partial T}$. Integrating over the Boltzmann distribution allows deriving specific heat in terms of energy variance:

$$C_T = \frac{\sigma^2(E(T))}{T^2} \tag{9}$$

where $\sigma^2(E(T))$ is the variance of the system energy at $T$.

### 5.2.3 Determination of Critical Temperature

To determine the critical temperature based on specific heat, we first denote $T(t)$ as the temperature at step $t$, $C(t)$ as the specific heat at temperature $T(t)$, and $x_t$ as the solution sampled at step $t$, then by selecting an appropriate sample size $M$, we define the approximation of $C(t)$ as :

$$\hat{C}(t) = \frac{\sigma^2(\{f(x_{t-M+1}), \cdots, f(x_t)\})}{T(t)^2}, \quad t \geq M \tag{10}$$

where $\sigma^2(\{f(x_{t-M+1}), \cdots, f(x_t)\})$ represents the variance in objective values over the $M$ most recent steps. The critical temperature $T_c$ can be determined as $T(t^*)$, where $t^* = \underset{t \geq M}{\arg\max}\, \hat{C}(t)$. However, SA with gradient-based discrete samplers convergences rapidly in the initial stage, resulting in an *abnormal* initial peak in specific heat (due to the high variance). As the annealing progresses, the specific heat quickly decreases, eventually stabilizing at a level more typical of a critical temperature. This behavior is shown in Figure 4a&4b.

To address the abnormal peak, we introduce a "skip step" threshold, denoted as $t_{\text{skip}}$, ensuring that the initial transient behavior is excluded from the analysis. Thus, the critical temperature is identified as $T(\tilde{t}^*)$, where

$$\tilde{t}^* = \underset{t \geq t_{\text{skip}}}{\arg\max}\, \hat{C}(t). \tag{11}$$

Our method diverges from traditional SA reheat strategies (Abramson et al., 1999) by accounting for inhomogeneous chains and addressing gradient-based methods' unique abnormal peaks.

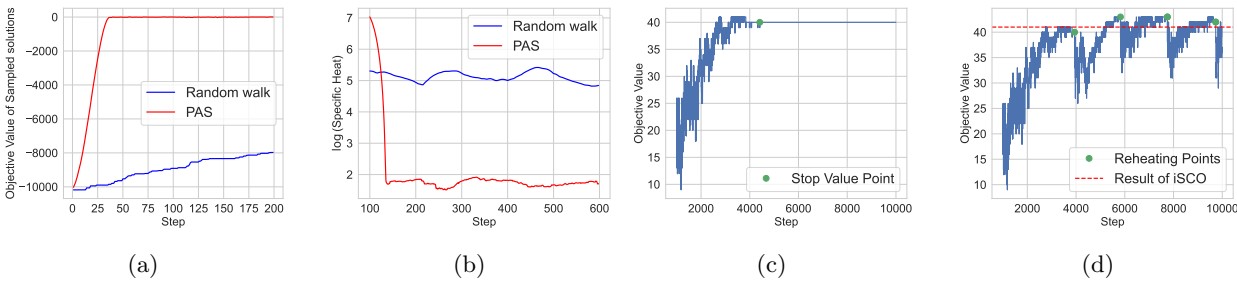

Figure 4: **(a):** Comparison of the first 200 steps shows that SA with PAS, a gradient-based discrete sampler, achieves fast initial improvement, then slows, unlike traditional SA's steady progress. **(b):** From steps 100 to 600, SA with PAS initially has much higher specific heat, which then quickly drops and stabilizes, in contrast to SA's steady specific heat levels. **(c):** After converging to the stop value point, iSCO begins to wander in contours and wastes the rest of the steps. **(d):** Upon detecting "wandering in contours", ReSCO reheats to the critical temperature, facilitating an escape from this behavior and discovering better solutions.

Table 1: **Left:** Average approximation ratio of different methods on two MaxClique tasks, where iSCO-N and ReSCO-N represent running N sampling chains for each problem; **Right:** Runtime of single-chain iSCO, single-chain ReSCO, and 32-chain iSCO on MIS tasks on a machine with a single NVIDIA RTX A6000 GPU.

| Method | Twitter | RBtest |
|---|---|---|
| EPM (Karalias & Loukas, 2020) | 0.924 | 0.788 |
| AFF (Wang et al., 2022) | 0.926 | 0.787 |
| RUN-CSP (Tönshoff et al., 2020) | 0.987 | 0.789 |
| iSCO-1 (Sun et al., 2023b) | 0.971 | 0.795 |
| iSCO-16 (Sun et al., 2023b) | 1.000 | 0.854 |
| ReSCO-1 (ours) | 0.972 | 0.811 |
| ReSCO-16 (ours) | **1.000** | **0.859** |

| Method | SATLIB | ER-[700-800] | ER-[9000-11000] |
|---|---|---|---|
| iSCO-1 | 40 min | 8 min | 193 min |
| ReSCO-1 | 69 min | 17 min | 234 min |
| iSCO-32 | 826 min | 225 min | 1746 min |

### 5.3 Reheated Gradient-based Discrete Sampling for Combinatorial Optimization

Combining results in Section 5.1 and 5.2, we obtain the ***Re****heated **S****ampling for **C****ombinatorial **O****ptimization* algorithm (ReSCO), which is summarized in Algorithm 1. ReSCO is compatible with any gradient-based discrete sampler. In this paper, we use the same sampler as in Sun et al. (2023b). ReSCO adds minimal overhead, as the reheat mechanism can be cheaply implemented, which is verified in the experiments section.

## 6 Experiments

We conduct a thorough empirical evaluation of our method ReSCO on various graph combinatorial optimization tasks, including Max Independent Set (MIS), MaxClique, MaxCut and Graph Balanced Partition. For all problems, we follow the experimental setup in Sun et al. (2023b).

We found that ReSCO requires minimal hyperparameter tuning. For Max Independent Set, MaxClique and MaxCut, we use a value threshold $\epsilon = 0.01$, a wandering length threshold $N = 100$, and a sample size $M = 100$ for all tasks. The skip step threshold $t_{\text{skip}}$ varies by problem and can be easily set at the point where specific heat's initial sharp decrease ends. See Appendix D for more experimental details, and see Appendix E for discussions about how to choose hyperparameters. For Graph Balanced Partition, the value threshold is adjusted per problem set, and the details are provided in Appendix H.

We present the results of experiments on Max Independent Set and MaxClique in this section, and put the problem description and dataset information in Appendix F. The problem description, dataset information and experimental results of MaxCut and Graph Balanced Partition are presented in Appendix G and H respectively.

---

**Algorithm 1** Reheated Sampling for Combinatorial Optimization

---

1: **Input:** temperature schedule $T(\cdot)$, sampling chain length $L$, value threshold $\epsilon$, wandering length threshold $N$, sample size $M$, skip step threshold $t_{\text{skip}}$
2: **Initialize:** state $x_0$, critical temperature $\tilde{t}^* = t_{\text{skip}}$, temperature indicator step $t_{\text{Temp}} = 1$, maximum specific heat $C^* = 0$, reheated = false
3: **for** $t = 1$ **to** $L$ **do**
4:      $x_t \leftarrow$ SA-gradient-based-Sample$(x_{t-1}, T(t_{\text{Temp}}))$
     ▷ *To determine critical temperature $\tilde{t}^*$*
5:      **if** $t \geq t_{\text{skip}}$ and not reheated **then**
6:          Calculate $\hat{C}(t)$ as in equation 10
7:          **if** $\hat{C}(t) \geq C^*$ **then**
8:              $C^* \leftarrow \hat{C}(t), \tilde{t}^* \leftarrow t$
9:          **end if**
10:      **end if**
     ▷ *To detect "wandering in contours"*
11:      **if** Condition equation 8 holds **then**
12:          $t_{\text{Temp}} \leftarrow \tilde{t}^*$    ▷ *Reheat to critical temperature*
13:      **end if**
14:      $t_{\text{Temp}} \leftarrow t_{\text{Temp}} + 1$
15: **end for**

---

Table 2: Results of ReSCO and recently proposed methods when solving MIS-ER-[700-800] problem set. ReSCO surpasses all of them and obtains the best result on this problem set to our knowledge.

| Method | ReSCO-32 (ours) | GFlowNets (Kim et al., 2024) | DIFUSCO (Sun & Yang, 2023) | T2T (Li et al., 2024) |
|---|---|---|---|---|
| Results | **45.24** | 41.14 | 40.35 | 41.37 |

## 6.1 Results and Analysis

### 6.1.1 Multiple Chains

We report the results of MIS and MaxClique in Table 3 and Table 1 (**Left**) respectively, with KaMIS serving as the baseline against which we measure performance drop. We see that ReSCO under the multi-chain setting surpassed all learning-based and sampling methods and even found better solutions than using optimization solvers in some tasks.

We also compare the results of ReSCO and other recently proposed methods which also run experiments on MIS-ER-[700-800], and the results are reported in Table 2. It shows that ReSCO surpasses all of the recently proposed methods and obtains the best result on this problem set to our knowledge.

### 6.1.2 Single Chain

When comparing the results between iSCO and ReSCO under multi-chain setting, the improvements of ReSCO may seem minor. This is because running multi-chains can mitigate the wandering in contours problem to some extent, but the extra computational cost compared to the single-chain setting is extremely large.

We report the runtime of single-chain iSCO, single-chain ReSCO and multi-chain (32-chains) iSCO when solving MIS problem sets in Table 1 (**Right**). The experiments were conducted on a single A6000 with multiple chains executed in parallel, using the same implementation used by the iSCO paper. We see that the runtime of 32-chain iSCO is at least 9 times of the runtime of single-chain iSCO, and the runtime of 32-chain iSCO is 28 times of the runtime of single-chain iSCO when solving MIS-ER-[700-800].

Table 3: Results of MIS on three benchmarks provided by Qiu et al. (2022). Baselines involve methods using optimization solvers (OR), Reinforcement Learning (RL), Supervised Learning (SL) equipped with Tree Search (TS), Greedy decoding (G), and Sampling (S). S-$N$ represents running $N$ chains for each problem. Methods that can't produce results in 10x time limit of DIMES are labeled as N/A.

| Method | Type | SATLIB | | ER-[700-800] | | ER-[9000-11000] | |
|---|---|---|---|---|---|---|---|
| | | Size ↑ | Drop ↓ | Size ↑ | Drop ↓ | Size ↑ | Drop ↓ |
| KaMIS | OR | 425.96* | - | 44.87* | - | 381.31* | - |
| Gurobi | OR | 425.95 | 0.00% | 41.38 | 7.78% | N/A | N/A |
| Intel (Li et al., 2018a) | SL+TS | N/A | N/A | 38.8 | 13.43% | N/A | N/A |
| | SL+G | 420.66 | 1.48% | 34.86 | 22.31% | 284.63 | 25.35% |
| DGL (Böther et al., 2022) | SL+TS | N/A | N/A | 37.26 | 16.96% | N/A | N/A |
| LwD(Ahn et al., 2020) | RL+S | 422.22 | 0.88% | 41.17 | 8.25% | 345.88 | 9.29% |
| DIMES(Qiu et al., 2022) | RL+G | 421.24 | 1.11% | 38.24 | 14.78% | 320.50 | 15.95% |
| | RL+S | 423.28 | 0.63% | 42.06 | 6.26% | 332.80 | 12.72% |
| iSCO (Sun et al., 2023b) | S-1 | 422.65 | 0.78% | 43.37 | 3.3% | 377.44 | 1.0% |
| | S-32 | 424.16 | 0.42% | 45.16 | -0.6% | 383.50 | -0.5% |
| ReSCO(Ours) | S-1 | 422.76 | 0.75% | 44.18 | 1.5% | 378.25 | 0.8% |
| | S-32 | **424.21** | **0.42%** | **45.24** | **-0.8%** | **383.75** | **-0.6%** |

Table 4: Results of Simulated Annealing with different gradient-based samplers with/without reheat using single-chain setting on MIS-ER-[700-800] problem set. Reheat improves the results for all the gradient-based samplers.

| Sampler | PAS (iSCO/ReSCO) | DMALA ((Zhang et al., 2022)) | DLMC ((Sun et al., 2023a)) |
|---|---|---|---|
| Without reheat | 43.37 | 42.92 | 42.94 |
| With reheat | **44.18** | **43.58** | **43.64** |

When we turn to the single-chain setting (which is the common setting in most algorithms), the improvement of ReSCO is significant. For example, under the single-chain setting, iSCO obtains 43.37 while ReSCO obtains 44.18 on MIS-ER-[700-800] problem set, and iSCO obtains 377.44 while ReSCO obtains 378.25 on MIS-ER-[9000-11000] problem set (see Table 3).

Also, single-chain ReSCO surpasses single-chain iSCO given the same runtime when solving MIS-ER-[700-800]. We compare the performance of single-chain iSCO and single-chain ReSCO given the same runtime when solving MIS-ER-[700-800], and report the results in Figure 5a. Though iSCO performed better in the first 200 seconds, it was surpassed by ReSCO in the remaining 800 seconds. Besides, iSCO showed a tendency to stay at that level while ReSCO still showed an increasing trend.

## 6.2 Effect and Applicability of Reheat Mechanism

We further study the effect of reheat by applying ReSCO and iSCO on a MIS problem. As illustrated in Figure 4c&4d, iSCO (4c) begins "wandering in contours" within 5,000 steps, failing to find improved solutions thereafter. In contrast, ReSCO (4d) manages to escape from "wandering in contours" each time after reheating (marked as green points) and identify better solutions, outperforming iSCO.

Furthermore, we test simulated annealing with three different gradient-based samplers with/without reheat mechanism on MIS-ER-[700-800] problem set with a single-chain setting to show the effect of reheat on different samplers. As can be seen in Table 4, reheat mechanism improves the results of all three gradient-based samplers, proving its efficacy and broad applicability.

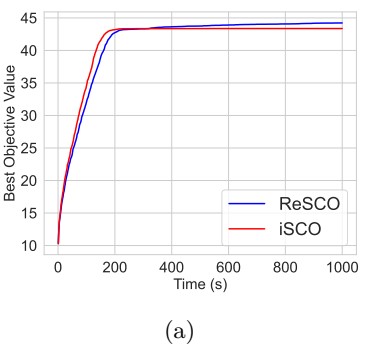 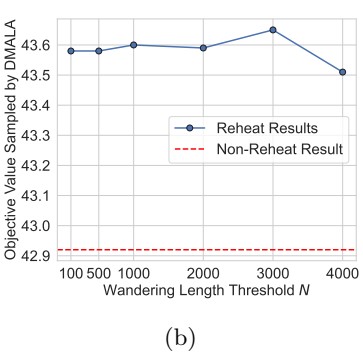 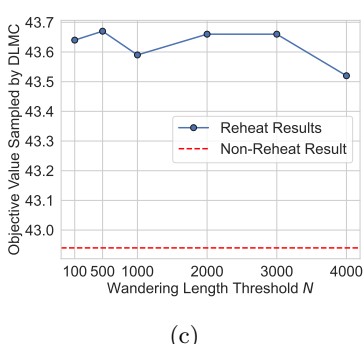

(a)        (b)        (c)

Figure 5: **(a)**: The best value obtained by single-chain iSCO and single-chain ReSCO within the same runtime when solving the MIS-ER-[700-800] problem set. **(b)**: Performance of Reheated DMALA under different wandering length threshold settings. **(c)**: Performance of Reheated DLMC under different wandering length threshold settings.

## 6.3 Ablation Study

### 6.3.1 Final Temperature

To evaluate ReSCO's robustness against variations in the cooling schedule, we conducted experiments on the RBtest dataset with various final temperatures (we did not consider initial temperature due to its demonstrated insensitivity (Sun et al., 2023b)).

We reported the results in Figure 6a, which indicate that ReSCO consistently outperforms iSCO on all final temperature values and is relatively robust to different values. In contrast, iSCO's performance declines noticeably with low final temperatures. This suggests that ReSCO's reheat mechanism enables it to be adaptive across a range of temperatures by reheating to the critical temperature each time the current temperature is too low for it to find better solutions.

### 6.3.2 Chain Length

With constant initial and final temperatures, extending the chain length and slowing the cooling rate generally improves results. We tested the impact of chain length on performance using the RBtest dataset under a single-chain setup, keeping initial and final temperatures at 1 and $10^{-3}$, respectively, and varying chain lengths from 1k to 40k. Results illustrated in Figure 6b show that extending the chain length boosts performance for both algorithms, with ReSCO consistently outperforming iSCO at varying chain lengths.

### 6.3.3 Wandering Length Threshold

Among the four new hyperparameters introduced in ReSCO $(\epsilon, M, t_{\text{step}}, N)$, determining the wandering length threshold $N$ poses the greatest challenge due to the varying nature of "wandering in contours" across different problems. In fact, setting $N$ too large reduces ReSCO back to iSCO. To assess $N$'s impact on ReSCO's performance, we performed experiments in a single-chain setting on the SATLIB and ER-[9000-11000] datasets with varying $N$ values, using iSCO's outcomes as a reference baseline.

As demonstrated in Figures 6c and 6d, ReSCO exhibits robust performance across varying wandering length thresholds. It consistently outperforms iSCO in finding better solutions for both datasets, regardless of the threshold chosen.

This claim is further substantiated by Figure 5b and 5c, where we implemented the reheat mechanism in other gradient-based discrete samplers: DMALA (Zhang et al., 2022) (Left) and DLMC (Sun et al., 2023a) (Right). We report the results of these two samplers under different wandering length threshold settings when solving MIS-ER-[700-800] problem set. Both reheated DMALA and DLMC outperformed their non-

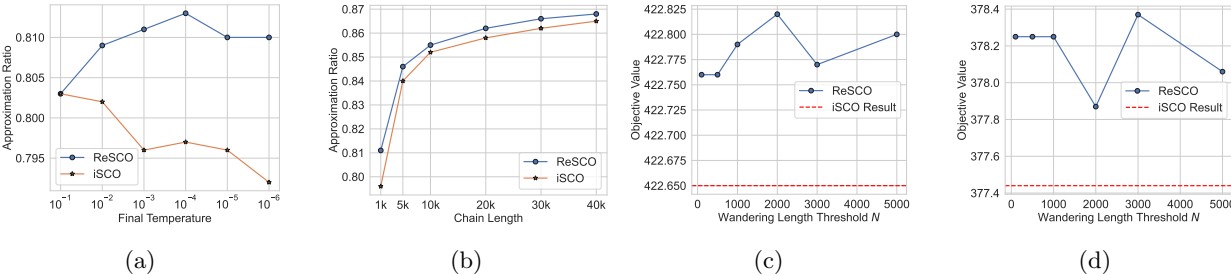

Figure 6: Ablation studies on final temperature (a), chain length (b), and wandering length threshold (c) & (d).

reheated counterparts across all threshold settings. These results align with those observed for ReSCO, further demonstrating the robustness and effectiveness of our reheat mechanism across different samplers.

However, we cannot find a specific pattern of how the wandering length threshold influences the results. We leave this for a future direction.

## 7 Conclusions and Limitations

In this paper, we identify a problematic behavior observed in gradient-based discrete samplers for combinatorial optimization, denoted as "wandering in contours". To counteract this issue, we introduce a novel reheat mechanism inspired by the principles of critical temperature and specific heat. This approach enables effective detection of this undesired behavior and resets the temperature to an optimal level that ensures a balance between exploration and exploitation. Our extensive experimental results validate the efficacy of the proposed method. However, there is still room for improvement. We list a few future directions below:

**Hyperparameter Tuning** Although ReSCO requires minimal hyperparameter tuning, further research on optimizing its parameters could improve its performance. For example, determining the wandering length threshold $N$ remains challenging, and developing a more systematic approach for setting this hyperparameter is an important direction for future work.

**Broader Applicability** While our reheating mechanism significantly enhances gradient-based discrete samplers for various combinatorial optimization (CO) problems, its reliance on gradient information limits its applicability to CO problems with non-differentiable objectives. A future research is to extend our method to non-differentiable CO problems and more general optimization tasks in the future.

**Theoretical Analysis** We have demonstrated the effectiveness of our reheat mechanism through extensive experiments. However, a theoretical analysis of the reheat mechanism is still lacking. Providing a theoretical foundation for our method will offer deeper insights into its behavior.

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

# A  Broad Existence of "Wandering in Contours"

To demonstrate the widespread occurrence of wandering in contours in gradient-based discrete samplers, we analyzed the trajectory of the objective values of DMALA (Zhang et al., 2022), as depicted in Figure 7.

Figure 7 (**Left**) reveals that DMALA begins to wander in contours after 4000 steps, and wastes about 3000 steps before it escapes from wandering in contours. However, it continues to wander in contours as soon as it escapes and wastes the rest steps obtaining no improvement. Figure 7 (**Right**) illustrates that the objective values continue to fluctuate but remain very similar to or identical to the stop value, indicating that DMALA is indeed wandering in contours after this point.

The behavior exhibited in Figure 7 closely resembles that shown in Figures 2b & 2c. This similarity suggests that despite DMALA's simultaneous update of all dimensions, which theoretically allows for more comprehensive exploration, it still exhibits wandering in contours nearly identical to PAS (Sun et al., 2021). This observation provides strong evidence that wandering in contours is a common phenomenon among gradient-based discrete samplers.

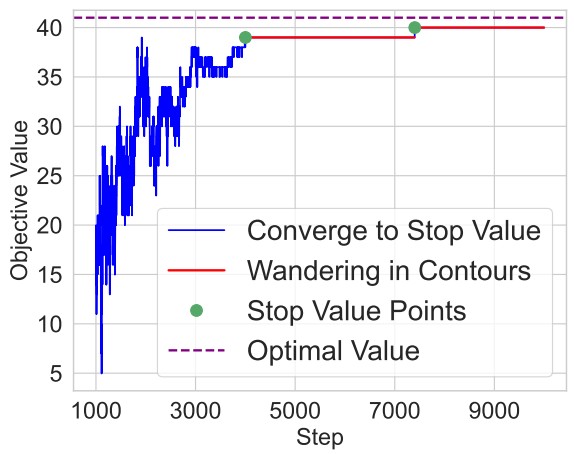 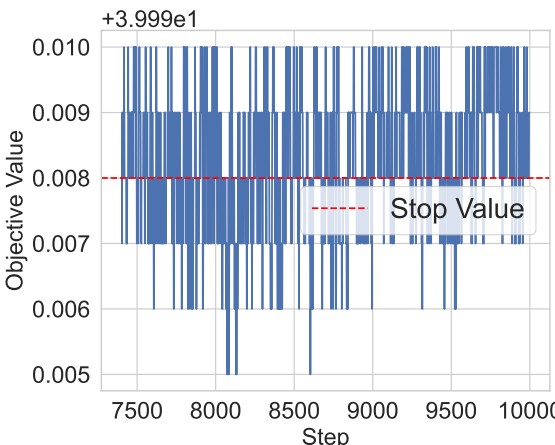

Figure 7: **Left:** The trajectory of objective values of DMALA, which is similar to Figure 2b, showing the existence of "Wandering in Contours". **Right:** A detailed view of objective value after DMALA wanders in contours. Its similarity with Figure 2c further proves the existence of "Wandering in Contours" in DMALA.

# B  Comparison of "Wandering in Contours" with Plateau Phenomenon

The "wandering in contours" behavior observed in gradient-based discrete sampling with simulated annealing resembles the *Plateau Phenomenon* Fukumizu & Amari (2000); Wei et al. (2008); Yoshida & Okada (2020), encountered in the training of deep learning models. In deep learning, a plateau refers to a phase where the training loss decreases at an exceedingly slow rate over many epochs. This is akin to "wandering in contours", where the algorithm repeatedly samples solutions around a stop value without making progress. This similarity highlights a pervasive challenge in gradient-based optimization algorithms, extending across both continuous and discrete domains. Addressing the plateau phenomenon typically involves modifying neural network architectures, an approach that does not translate to the discrete sampling problems discussed in this paper.

# C  Comparison of Reheating with Stepsize-tuning Methods

Reheating mechanism is fundamentally different from existing stepsize-tuning methods explored in MCMC literature (Zhang et al., 2019; Andrieu & Thoms, 2008). Stepsize tuning methods, such as adaptive stepsize (Andrieu & Thoms, 2008) or cyclical stepsize (Zhang et al., 2019), control how far the next state will

be from the current state without changing the stationary distribution. In contrast, reheating mechanism changes the stationary distribution through temperature adjustment.

In fact, stepsize-tuning methods cannot effectively mitigate the problem of wandering in contours. To illustrate this, consider the update rule of the DMALA (Zhang et al., 2022):

$$p(x_i'|x) = \text{Softmax}\left(\frac{1}{2}\frac{\nabla f(x)_i(x_i'-x_i)}{T} - \frac{(x_i'-x_i)^2}{2\alpha}\right)$$

When wandering in contours occurs, the temperature $T$ inside the algorithm tends to be very small or even close to 0. In this case, regardless of how the stepsize $\alpha$ is tuned, the gradient term remains dominant in the update rule. Consequently, the misleading gradient information perpetuates the wandering in contours behavior.

To empirically validate this, we conducted experiments using the iSCO algorithm on three MIS problems. Upon detecting wandering in contours, we adjusted the stepsize $\alpha_1$ to larger values $\alpha_2$ and recorded the final ratios (iSCO result divided by baseline result) as shown in Table 5.

Table 5: Effect of Stepsize-tuning on solving Wandering in Contours

| $\alpha_1$ | $\alpha_2$ | Problem 1 | Problem 2 | Problem 3 |
|---|---|---|---|---|
| 0.2 | 0.2 | 0.79 | 0.83 | 0.83 |
| 0.2 | 1 | 0.79 | 0.83 | 0.83 |
| 0.2 | 10 | 0.79 | 0.83 | 0.83 |
| 0.2 | $10^5$ | 0.79 | 0.83 | 0.83 |
| 0.2 | $\infty$ | 0.79 | 0.83 | 0.83 |

As evident from Table 5, increasing the stepsize when wandering in contours is detected does not improve performance across all three problems. This empirical evidence supports our theoretical argument that stepsize-tuning is ineffective in addressing the wandering in contours issue.

## D   Settings of Experiments

We report the settings of all the experiments in Table 6.

Table 6: Setting of hyper-parameters in the experiments. ER1 represents ER-[700-800] problem set, and ER2 represents ER-[9000-11000] problem set.

| Datasets | MIS | | | MaxClique | | MaxCut | |
|---|---|---|---|---|---|---|---|
| | SATLIB | ER1 | ER2 | Twitter | RBtest | Optsicom | BA |
| Initial Temperature $T(0)$ | 1.0 | 1.0 | 1.0 | 1.0 | 1.0 | 1.0 | 1.0 |
| Ending Temperature $T(L)$ | 1e-5 | 1e-3 | 1e-3 | 1e-2 | 1e-3 | 1e-3 | 1e-6 |
| Chain Length $L$ | 1M | 200k | 400k | 1k | 1k | 25k | 50k |
| Temperature Schedule $T(\cdot)$ | exponentially decay | | | | | | |
| Sample Size $M$ | 100 | 100 | 100 | 100 | 100 | 100 | 100 |
| Value Threshold $\epsilon$ | 0.01 | 0.01 | 0.01 | 0.01 | 0.01 | 0.01 | 0.01 |
| Wandering Length Threshold $N$ | 100 | 100 | 100 | 100 | 100 | 100 | 100 |
| Skip Step $t_{skip}$ | 500 | 200 | 350 | 100 | 130 | 100 | 200 |

## E   Determination of Hyperparameters

Beyond the experiments conducted in this paper, we give some methods to choose the four hyperparameters in ReSCO here.

The value threshold $\epsilon$ should be a small number that can cover the value fluctuations caused by wandering in contours. A practical heuristic we found is setting $\epsilon$ to be 10 times the minimal increment in objective values. This approach has proven effective in our experiments for identifying wandering in contours, where $\epsilon$ was consistently set at 0.01 across all experiments.

The sample size $M$ can be subjectively selected within a certain range without significantly affecting the reheating mechanism.

The skip step $t_{\text{step}}$ is easily identified at the point where the specific heat ends its initial sharp decline.

The wandering length threshold $N$ is a bit more difficult to decide. As discussed in Section 6.3.3, the only requirement for $N$ is that it should not be too small (no smaller than 100), otherwise reheat will happen too often. And in practice, $N$ will not influence the final results too much. Figure 6c & 6d prove it: for all the chosen numbers varying from 100 and 5000, the results of ReSCO outperform the results of iSCO. In fact, iSCO can be seen as a special case of ReSCO with an extremely large $N$ (i.e. larger than the chain length), in which case reheat will never happen.

# F    Problem Setup for MIS and MaxClique

## F.1    Max Independent Set

**Problem Description**    Given a graph $G = (V, E)$ where $|V| = d$, the Max Independent Set (MIS) problem aims to find the largest set of vertices no two of which are adjacent. Representing the inclusion of node $i$ in the set as $x_i = 1$ and exclusion as $x_i = 0$, the MIS problem can be framed as the optimization problem:

$$\max_{x \in \{0,1\}^d} \sum_{i=1}^{d} x_i$$

subject to:

$$x_i x_j = 0, \quad \forall (i, j) \in E$$

Denote the adjacency matrix of $G$ as $A$, by selecting a penalty term $\lambda$, we can construct the energy function as:

$$f(x) = -\sum_{i=1}^{d} x_i + \lambda \frac{x^T A x}{2}$$

We choose $\lambda = 1.0001$, which is the same as Sun et al. (2023b), and report $-f(x)$ as the final result.

**Datasets**    Following Sun et al. (2023b), we use the MIS benchmark from Qiu et al. (2022), consisting of one realistic dataset SATLIB (Hoos & Stutzle, 2000) and ErdsRényi(ER) random graphs of different sizes. We directly test ReSCO on the test datasets provided by Goshvadi et al. (2023), which contains the following datasets:

- SATLIB: consists of 500 test graphs, each with at most 1347 nodes and 5978 edges.

- ER-[700-800]: consists of 128 test graphs with 700 to 800 nodes.

- ER-[9000-11000]: consists of 16 test graphs with 9000 to 11000 nodes.

## F.2    Max Clique

**Problem Description**    Given a graph $G = (V, E)$ with $|V| = d$, the Max Clique (MC) problem seeks the largest subset of vertices such that every two distinct vertices are adjacent. Using binary representation where $x_i = 1$ indicates node $i$ is part of the clique and $x_i = 0$ otherwise, the MC problem can be defined as:

$$\max_{x \in \{0,1\}^d} \sum_{i=1}^{d} x_i$$

subject to:

$$x_i x_j = 0, \quad \forall (i,j) \notin E$$

Letting $A'$ be the adjacency matrix of the graph complement, we can establish the energy function:

$$f(x) = -\sum_{i=1}^{d} x_i + \lambda \frac{x^T A' x}{2}$$

To align with prior work, we set $\lambda = 1.0$ as in Sun et al. (2023b), and the result reported is $-f(x)$.

**Datasets**   Following Sun et al. (2023b), we test ReSCO on:

- RB: synthetic datasets generated with RB model (Xu et al., 2007), consists of 500 graphs, each with at most 475 nodes;

- Twitter: a realistic Twitter dataset (Leskovec & Krevl, 2014), consists of 196 graphs, each with at most 247 nodes.

## G   Experimental Information on MaxCut

**Problem Description**   Given an undirected graph $G = (V, E)$ where $|V| = d$, the MaxCut problem aims to partition $V$ into two disjoint subsets such that the number of edges between the subsets is maximized. Each vertex $i$ is assigned to a partition represented by $x_i \in \{0,1\}$, where $x_i = 1$ indicates that vertex $i$ is in one subset and $x_i = 0$ indicates it's in the other. Then MaxCut problem can be formulated as:

$$\max_{x \in \{0,1\}^d} \sum_{(i,j) \in E} \frac{1 - (2x_i - 1)(2x_j - 1)}{2}$$

We can directly construct the energy function due to MaxCut is unconstrained, which is:

$$f(x) = - \sum_{(i,j) \in E} \frac{1 - (2x_i - 1)(2x_j - 1)}{2}$$

and we report $-f(x)$ ad the final result.

**Datasets**   Following Sun et al. (2023b), we apply the benchmark as in (Dai et al., 2020) and (Karalias & Loukas, 2020) and , and choosing the following datasets:

- BA: synthetic datasets generated with BarabásiAlbert model, consisting of 1000 graphs, with nodes number varying from 16 to 1000;

- Optsicom: realistic datasets consisting of ten graphs, each with at most 125 nodes.

Table 7: The improvement of approximation ratios compared to iSCO on MaxCut Problems; The second line of BA graphs represents the nodes number of graphs in the dataset.

| Methods | BA graphs | | | | | | | Optsicom |
|---|---|---|---|---|---|---|---|---|
| | 16-20 | 32-40 | 64-75 | 128-150 | 256-300 | 512-600 | 1024-1100 | |
| ReSCO | +0 | +2e-6 | +0 | +0 | +1e-4 | +1e-4 | +1e-5 | +0 |

**Results** Because iSCO has been shown to surpass all the learning-based methods and classical methods in baselines and obtained even better results compared to optimization solvers, so we only report the improvement of approximation ratios of ReSCO compared to iSCO with the single-chain setting on the two datasets in Table 7:

It's clear that the results obtained by iSCO and ReSCO are almost the same, and ReSCO surpasses iSCO slightly on several datasets. This is because iSCO has found solutions that are good enough, making exploring other parts of the solution space less useful.

## H  Experimental Information on Graph Balanced Partition

**Problem Description and Datasets** The Graph Balanced Partition problem aims to divide the nodes of a given graph $G = (V, E)$ into $K$ disjoint subsets $S_1, \ldots, S_k$, while simultaneously minimizing the number of edges that run between different partitions and ensuring that the partitions remain balanced in size. We apply the same settings and implementations as in (Goshvadi et al., 2023), and compare the results of ReSCO-1 (ours) and iSCO-1 (Sun et al., 2023b) using the following three metrics:

- **Objective Value:** Computed based on the edge-cut ratio and balanceness, as described in (Goshvadi et al., 2023). This metric reflects both the efficiency in minimizing the cut edges and the degree of balance among partitions.

- **Edge Cut Ratio:** Measures the proportion of edges in the graph that are cut between different partitions. A lower edge cut ratio indicates fewer inter-partition connections, which is desirable.

- **Balanceness:** Quantifies how evenly the nodes are distributed across the partitions. A high balanceness indicates that the partitions are of similar size, adhering to the balance constraint.

We compare the results of ReSCO-1 and iSCO-1 on five widely used TensorFlow graphs: VGG, MNIST-conv, ResNet, AlexNet, and Inception-v3. Detailed information regarding these graphs can be found in (Nazi et al., 2019).

**Hyperparameter Setting** For all the Graph Balanced Partition experiments, our setup is the same as in (Sun et al., 2023b). For the additional hyperparameters in ReSCO-1, we set the skip step $t_{skip}$ to 200k, and the wandering length threshold $N$ to 1k. The value threshold $\epsilon$ is adjusted per problem set, which is reported in Table 8.

Table 8: Value Thresholds for Different Graph Balanced Partition problem sets.

| Problem Sets | VGG | MNIST-conv | ResNet | AlexNet | Inception-v3 |
|---|---|---|---|---|---|
| Value Threshold | 0.5 | 0.5 | 1 | 0.5 | 2 |

**Results** We report the Objective Value, Edge cut ratio, and Balanceness of solutions found by ReSCO-1 and iSCO-1 on the five TensorFlow graphs in Table 9.

As shown, ReSCO-1 consistently outperforms iSCO-1 across all five tasks in terms of Objective Value, the primary metric directly optimized in the experiments. This demonstrates the effectiveness of our proposed

Table 9: Performance of ReSCO-1 and iSCO-1 on Graph Balanced Partition problems.

| Metric | Method | VGG | MNIST-conv | ResNet | AlexNet | Inception-v3 |
|---|---|---|---|---|---|---|
| Objective Value ↑ | iSCO-1 | -261.86 | -68.52 | -6483.95 | -104.60 | -4565.87 |
| | ReSCO-1 | **-261.28** | **-53.67** | **-6449.40** | **-72.18** | **-4563.21** |
| Edge cut ratio ↓ | iSCO-1 | **0.058** | 0.051 | 0.100 | 0.041 | 0.056 |
| | ReSCO-1 | 0.059 | **0.025** | **0.099** | **0.025** | **0.055** |
| Balanceness ↑ | iSCO-1 | 0.95 | **0.92** | 0.99 | **0.97** | 0.99 |
| | ReSCO-1 | **0.97** | 0.62 | 0.99 | 0.94 | 0.99 |

reheat mechanism. Although Edge Cut Ratio and Balanceness are not explicitly optimized by the objective, ReSCO-1 still achieves a better Edge Cut Ratio in four out of five tasks. For Balanceness, the results obtained by ReSCO-1 are competitive in most problem sets.

