# OpenReview forum: "Reheated Gradient-based Discrete Sampling for Combinatorial Optimization"
_TMLR — Accepted by TMLR_

### Review · Reviewer_zis4 · 2024-11-26

**Summary Of Contributions:**

This paper proposes a novel reheating mechanism to address the inefficiencies of gradient-based discrete sampling methods for combinatorial optimization (CO). It identifies a critical issue termed "wandering in contours," where the algorithm gets stuck exploring solutions with similar objective values. The paper introduces a reheating mechanism inspired by physics concepts such as critical temperature and specific heat, which facilitates escaping from these suboptimal contours. The proposed approach shows significant empirical improvements over existing sampling-based and data-driven CO methods on Max Independent Set and MaxClique.

The contributions of this paper are clear and practical, and the proposed reheating mechanism adds a valuable layer of robustness to existing sampling approaches for CO. However, while the empirical results are promising, the work lacks theoretical guarantees, such as the convergence analysis. Adding such theoretical guarantees would strengthen the impact and provide a deeper understanding of the effectiveness of the proposed reheat method. Overall, the paper presents an interesting and useful enhancement to discrete sampling techniques and is a solid contribution to the field.

**Audience:**

Yes

**Claims And Evidence:**

Yes

**Requested Changes:**

Adding theoretical guarantees would strengthen the impact and provide a deeper understanding of the effectiveness of the proposed reheat method.

**Strengths And Weaknesses:**

Strengths:
1) The paper proposes a novel reheating mechanism that addresses a key inefficiency in gradient-based discrete sampling, enhancing solution quality in combinatorial optimization.
2) Strong empirical results are provided, showing significant improvements across multiple benchmark problems.
3) The motivation is clearly stated, and the practical impact of the reheating mechanism is well-supported by the numerical experiments.

Weaknesses: Theoretical guarantees, such as convergence proofs for the simulated annealing framework, are lacking.

---

> ### Author Response · Authors · 2025-01-10
>
> Thanks for your valuable comments! Please find our responses to the questions below.
>
>
> **Q1:Adding theoretical guarantees would strengthen the impact and provide a deeper understanding of the effectiveness of the proposed reheat method.**
>
>
> A1: Thank you for the suggestion! We want to emphasize that our work has already made several contributions: discovering a new phenomenon for gradient-based samplers, analyzing its causes, developing a practical and effective algorithm to improve the performance of gradient-based samplers, and conducting extensive experimental analysis. While we did not provide theoretical analysis, extensive experimental results shown in the paper have shown the effectiveness of the proposed algorithm.
>
> We acknowledge that including a theoretical analysis would be valuable. However, no theoretical analysis of the reheating mechanism of simulated annealing has been proposed in the existing literature. Previous works on simulated annealing for combinatorial optimization, such as [1], also lack theoretical analysis. This is because developing a theoretical framework for simulated annealing and the reheated mechanism is inherently challenging and requires significant advancements in theoretical tools. Therefore we leave this for future work.
>
>
> References:
>
>
> [1]. Sun, Haoran, et al. "Revisiting sampling for combinatorial optimization." International Conference on Machine Learning. PMLR, 2023.

---

### Review · Reviewer_j436 · 2024-12-19

**Summary Of Contributions:**

This manuscript reports the "wondering in contours" phenomenon in combinatorial optimization and  proposes the reheated gradient-based discrete sampling to tackle the phenomenon. The main contribution is the combination of gradient-based sampling and reheating, which enhances the exploration over the search space.

**Audience:**

Yes

**Claims And Evidence:**

Yes

**Requested Changes:**

1. Regarding Weakness 1, please refine the analysis of  "wondering in contours". A suggestion is empirically investigating whether arbitrary gradient-based method (not only SA) can wonder in contours. If so, "wondering in contours" comes from the problem landscape, rather than caused by misleading gradient.
2. Regarding Weakness 2, please correct the explaination of low temperature environment of SA.

**Strengths And Weaknesses:**

Strengths:
1. The manuscript is well-structured and well-written. The experimental study is comprehensive.
2. The combination of gradient-based sampling and reheating boosts the overall performance, which is a meaningful improvement in the area.

Weakness:
1. Section 4.2, the analysis of what causes "wondering in contours" is not strongly convincing. Some "wondering in contours" phenomenons may be caused by misleading gradient information, but the main reason of such phenomenon could come from the problem itself: combinatorial problems' landscape is often multi-modal, that is, multiple solutions can have the same (or very similar) objective values. Gradient-based methods are very likely  "wonder in contours" over this landscape.
2. Section 4.2, The explaination of low temperature environment of simulated annealing (SA) is questionable.  At the low temperature stage, SA tends to exploit a neighbor region, resulting in falling into a local optima, rahter than forwarding along a road and getting farther away from the beginning solution (i.e., wondering in contours).

---

> ### Author Response · Authors · 2025-01-10
>
> Thanks for your valuable comments! Please find our responses to the questions below.
>
>
>
> **Q1: ‘wandering in contours’ may mainly caused by the multi-modal landscape of CO problems instead of ‘misleading gradient information’. A suggestion is empirically investigating whether arbitrary gradient-based method (not only SA) can wander in contours. If so, "wandering in contours" comes from the problem landscape, rather than caused by misleading gradient.**
>
>
> A1: Thank you for the suggestion! We believe that ‘wandering in contours’ is primarily caused by ‘misleading gradient information,’ rather than the multi-modal landscape of CO problems, for several reasons:
>
> - If ‘wandering in contours’ was due to the multi-modal landscape of CO problems, we would expect to observe it with non-gradient-based samplers as well. However, as shown in Fig. 2a, non-gradient-based samplers do not exhibit this behavior. The fact that ‘wandering in contours’ is observed exclusively with gradient-based samplers strongly suggests that the gradient information is the root cause.
>
> - While CO problems often have multiple solutions, these solutions are typically far apart [4]. This makes it unlikely to cause a sampler to continuously sample distinct solutions with very similar objective values as what happens in ‘wandering in contours’.
>
> Regarding the suggestion, we would like to clarify that **gradients are not defined in discrete spaces**; instead, they are computed based on a continuous relaxation of the discrete CO function, as detailed in Sec. 3.3. Consequently, all gradient-based methods for solving CO problems share the same issue of ‘misleading gradient information.’ Therefore, **we cannot decide the main reason for ‘wandering in contours’ by investigating whether the arbitrary gradient-based method also wanders in contours.**
>
>
>
> **Q2: Section 4.2, The explaination of low temperature environment of simulated annealing (SA) is questionable. At the low temperature stage, SA tends to exploit a neighbor region, resulting in falling into a local optima, rahter than forwarding along a road and getting farther away from the beginning solution (i.e., wondering in contours).**
>
>
>
> A2: At low temperatures, the samplers can still accept new solutions with very similar objective values, allowing the algorithm to continue progressing along a path of solutions with similar objective values. In this scenario, the sampler does not become trapped in a local region but instead keeps moving forward.
>
> For an explanation of why gradient-based discrete samplers exhibit ‘wandering in contours’ rather than becoming trapped in local optima, please refer to Section 4.2. In brief, in discrete spaces, the local minima defined by gradients—corresponding to local minima in the natural relaxation of the original problem—are often infeasible (i.e. not discrete). In high-dimensional discrete spaces, numerous feasible solutions tend to surround these infeasible local minima. The gradient information guides the algorithm to jump among these nearby feasible solutions, causing the sampled solutions to share similar objective values while differing from one another.
>
> We mention low temperature because **it lowers the possibility of the algorithm jumping out of ‘wandering in contours’**. Misleading gradient information keeps the algorithm moving among solutions near the infeasible local minimum, and lower temperatures further decrease the chance of breaking out of this behavior.
>
>
>
> References:
>
>
> [1]. Sun, Haoran, et al. "Revisiting sampling for combinatorial optimization." International Conference on Machine Learning. PMLR, 2023.
>
>
> [2]. Goshvadi, Katayoon, et al. "DISCS: a benchmark for discrete sampling." Advances in Neural Information Processing Systems 36 (2024).
>
>
> [3]. Sun, Zhiqing, and Yiming Yang. "Difusco: Graph-based diffusion solvers for combinatorial optimization." Advances in Neural Information Processing Systems 36 (2023): 3706-3731.
>
>
> [4]. Tayarani-N, Mohammad-H., and Adam Prügel-Bennett. "On the landscape of combinatorial optimization problems." IEEE Transactions on Evolutionary Computation 18.3 (2013): 420-434.

---

> > ### Comment · Reviewer_j436 · 2025-01-21
> > **ACK**
> >
> > Thanks for the response. The response sounds good to me.

---

### Review · Reviewer_jKxQ · 2024-12-27

**Summary Of Contributions:**

This paper presents a simulated annealing method using gradient-based sampling for combinatorial optimization. Specifically, the authors point out the problem, termed wandering in contours, observed in existing methods and introduce the reheating technique into gradient-based sampling methods to improve the search efficiency. The experimental evaluation shows that the proposed method outperforms existing gradient sampling-based simulated annealing methods and data-driven algorithms.

**Audience:**

Yes

**Broader Impact Concerns:**

I do not have concerns about the ethical implications of this paper.

**Claims And Evidence:**

Yes

**Requested Changes:**

- As the experimental evaluation in this paper seems to be limited, it would be better to discuss the performance of the proposed method on various combinatorial optimization problems. Particularly, the performances on the balanced graph partition and traveling salesman problems treated in the previous paper [Sun et al. 2023b] will be useful.
- [Minor comment] To apply the gradient-based discrete sampling to combinatorial optimization, the objective function $u$ and constraint function $v$ should be differentiable with respect to solution $x$. Therefore, problems to which gradient-based discrete sampling methods can be applied are limited compared to ordinary simulated annealing. The limitation of the proposed method and gradient-based sampling for combinatorial optimization should be clarified.

**Strengths And Weaknesses:**

[Strengths]
- The problem of wandering in contours in gradient-based sampling for combinatorial optimization is pointed out.
- A solution for wandering in contours problem, a reheating mechanism based on the critical temperature, is proposed.

[Weaknesses]
- The proposed reheating technique seems somewhat straightforward. The theoretical justification of its algorithm design is unclear. For example, the detection of wandering in contours is affected by the scale of the objective function and is not invariant to the monotonically increasing transformation of $f$.
- The experimental evaluation might be limited. The previous work [Sun et al. 2023b] also tackles the balanced graph partition and traveling salesman problems, whereas this paper only treats MIS, MaxClique, and MaxCut.

---

> ### Author Response · Authors · 2025-01-10
>
> Thanks for your valuable comments! Please find our responses to the questions below.
>
>
> **Q1: As the experimental evaluation in this paper seems to be limited, it would be better to discuss the performance of the proposed method on various combinatorial optimization problems.**
>
>
> A1: Thanks for pointing out! We’re running experiments on the balanced graph partition problem sets and will post the results here and add them to the paper as soon as all experiments are finished.
>
> The previous paper [5] implemented the traveling salesman problem using non-gradient-based discrete samplers due to its non-differentiable objective. For this reason, we did not include them in the paper, as our focus is on gradient-based samplers. The phenomenon of ‘wandering in contours’ rarely occurs when using non-gradient-based discrete samplers since they converge slowly as shown in Fig2a in the paper. **Our proposed reheating mechanism primarily targets gradient-based discrete samplers**.
>
>
>
>
>
> **Q2: Problems to which gradient-based discrete sampling methods can be applied are limited compared to ordinary simulated annealing. The limitation of the proposed method and gradient-based sampling for combinatorial optimization should be clarified.**
>
>
> A2: Thank you for your suggestion! We agree that gradient-based methods have limitations when applied to nondifferentiable CO problems and will include a discussion on this in the revised version. However, we would like to emphasize that our proposed reheating mechanism remains valuable for several reasons.
>
> To begin with, **many CO problems have differentiable objective functions**, and gradient-based methods can be applied to them, such as. MIS, MaxCut, Graph Partition, and Knapsack Problem. For such problems, **gradient-based discrete samplers perform much better than ordinary simulated annealing [5, 6]**, and our reheating mechanism further improves gradient-based discrete samplers by alleviating ‘wandering in contours’.
>
> Moreover, even for **CO problems with nondifferentiable objective functions, zero-order approximation can be used** to make gradient-based discrete samplers applicable, as shown in [3]. While we focus on gradient-based discrete samplers because they are especially susceptible to “wandering in contours” (see Section 4.2 of the paper), and thus benefit greatly from our reheating mechanism, the same mechanism is also compatible with non-gradient-based samplers.
>
>
>
>
> References:
>
>
> [1]. Giacomo Zanella. Informed proposals for local mcmc in discrete spaces. Journal of the American Statistical Association, 115(530):852–865, Apr 2019. doi: 10.1080/01621459.2019.1585255.
>
>
> [2]. Will Grathwohl, Kevin Swersky, Milad Hashemi, David Duvenaud, and Chris Maddison. Oops i took a gradient: Scalable sampling for discrete distributions. In International Conference on Machine Learning, pp. 3831–3841. PMLR, 2021.
>
>
> [3]. Xiang, Y., Zhu, D., Lei, B., Xu, D., and Zhang, R. Efficient informed proposals for discrete distributions via newton’s series approximation. In International Conference on Artificial Intelligence and Statistics, pp. 7288–7310. PMLR, 2023.
>
>
> [4].Haoran Sun, Hanjun Dai, Wei Xia, and Arun Ramamurthy. Path auxiliary proposal for mcmc in discrete space. In International Conference on Learning Representations, 2021.
>
>
> [5]. Sun, Haoran, et al. "Revisiting sampling for combinatorial optimization." International Conference on Machine Learning. PMLR, 2023.
>
>
> [6]. Goshvadi, Katayoon, et al. "DISCS: a benchmark for discrete sampling." Advances in Neural Information Processing Systems 36 (2024).

---

> > ### Comment · Reviewer_jKxQ · 2025-01-22
> >
> > Thank you for the response. The response is reasonable for me. As I wanted to know the performance of the proposed method on other problems, please let me know if you added the experiments on balanced graph partition problem sets.

---

> > > ### Author Response · Authors · 2025-01-25
> > >
> > > Thank you for your response! We present the results of the Balanced Graph Partition problems in the table below, comparing the performance of the best solutions obtained by iSCO [1] and ReSCO (ours) under the 1-chain setting:
> > >
> > >
> > >
> > > | Metric             |  Method |     VGG     | MNIST-conv |    ResNet    |   AlexNet  | Inception-v3 |
> > > |--------------------|:-------:|:-----------:|:----------:|:------------:|:----------:|:------------:|
> > > |  Objective Value ↑ |  iSCO-1 |   -261.86   |   -68.52   |   -6483.95   |   -104.60  |   -4565.87   |
> > > |                    | ReSCO-1 | **-261.28** | **-53.67** | **-6449.40** | **-72.18** | **-4563.21** |
> > > |  Edge cut ratio ↓  |  iSCO-1 |    **0.058**    |    0.051   |     0.100    |    0.041   |     0.056    |
> > > |                    | ReSCO-1 |    0.059    |    **0.025**   |    **0.099**    |   **0.025**  |    **0.055**    |
> > > |  Balanceness ↑     |  iSCO-1 |     0.95    |    **0.92**    |     0.99     |    **0.97**    |     0.99     |
> > > |                    | ReSCO-1 |     **0.97**    |    0.62    |     0.99     |    0.94    |     0.99     |
> > >
> > >
> > >
> > >
> > > In the table, we compare iSCO and ReSCO through three metrics:
> > >
> > > - Objective value: It is computed based on the edge-cut ratio and balanceness in the same manner as described in [1].
> > >
> > > - Edge cut ratio: It measures the proportion of edges in the graph that are cut between different partitions.
> > >
> > > - Balanceness: It measures how evenly the nodes are distributed across the partitions.
> > >
> > >
> > >
> > > As shown, **ReSCO consistently outperforms iSCO across all five tasks in terms of Objective Value, the primary metric directly optimized in the experiments**. This demonstrates the effectiveness of our proposed reheat mechanism.
> > >
> > >
> > > Although **Edge Cut Ratio** and **Balanceness** are not explicitly optimized by the objective, ReSCO still achieves a better Edge Cut Ratio in four out of five tasks. For Balanceness, the results obtained by ReSCO are competitive in most problem sets.
> > >
> > >
> > > Our experimental setup is the same as in [1]. For the additional hyperparameters in ReSCO-1, we set the skip step to 200k, and the wandering length threshold to 1k for all the experiments. The value threshold is adjusted per problem set, which is reported in the following table:
> > >
> > >
> > >
> > > |  |     VGG     | MNIST-conv |    ResNet    |   AlexNet  | Inception-v3 |
> > > |:-------:|:-----------:|:----------:|:------------:|:----------:|:------------:|
> > > |  Value Threshold|  0.5   |   0.5  |   1  |   0.5 |   2   |
> > >
> > >
> > >
> > >
> > > References:
> > >
> > >
> > >
> > > [1]. Sun, Haoran, et al. "Revisiting sampling for combinatorial optimization." International Conference on Machine Learning. PMLR, 2023.

---

> > > > ### Comment · Reviewer_jKxQ · 2025-02-07
> > > >
> > > > Thank you for the additional experimental result. It will make the effectiveness of the proposed method clearer. I am satisfied with the authors' response. Thanks again.
> > > >
> > > > Best,

---

### Decision · Action_Editor_oBqF · 2025-02-25

**Recommendation:** Accept with minor revision

**Comment:**

All reviewers recommend acceptance.

I am marking this as "accept with minor revision" to allow the authors to incorporate changes addressing the reviewer comments into the paper, especially regarding the comments of Reviewers jKxQ and j436, as this was not done during the discussion phase.

Please note that TMLR does not have a strict page limit, so you can just add the results of the new experiments in the main text.

**Audience:**

All reviewers agree that the paper would be interesting to researchers working on gradient-based combinatorial optimisation.

**Claims And Evidence:**

All reviewers agree that the submission is supported by accurate, convincing and clear evidence.

---

> ### Author Response · Authors · 2025-03-06
>
> Dear AE,
>
>
> Thank you very much for your decision and for the valuable advice provided. We have carefully revised our paper according to all the reviewers’ suggestions and have uploaded the camera-ready version.
>
>
> In this final version, we have made the following updates:
> 1. Added the experimental results and the framework for Graph Balanced Partition to Section 6 and Appendix H.
> 2. Expanded Section 7 with a discussion on potential limitations and future directions, including the concern raised by Reviewer jKxQ.
> 3. Included the GitHub repository link for the paper.
>
>
> Once again, we appreciate your support and all the reviewers’ insightful comments.
>
>
>
> Sincerely,
>
>
> Authors